# Role of 3-Mercaptopyruvate Sulfurtransferase (3-MST) in Physiology and Disease

**DOI:** 10.3390/antiox12030603

**Published:** 2023-03-01

**Authors:** Swetha Pavani Rao, Prakashkumar Dobariya, Harshini Bellamkonda, Swati S. More

**Affiliations:** Center for Drug Design, College of Pharmacy, University of Minnesota, Minneapolis, MN 55455, USA

**Keywords:** 3-mercaptopyruvate sulfurtransferase, hydrogen sulfide, neurodegenerative disease, 3-mercaptopyruvate, sulfanegen, cyanide poisoning, therapeutics

## Abstract

3-mercaptopyruvate sulfurtransferase (3-MST) plays the important role of producing hydrogen sulfide. Conserved from bacteria to Mammalia, this enzyme is localized in mitochondria as well as the cytoplasm. 3-MST mediates the reaction of 3-mercaptopyruvate with dihydrolipoic acid and thioredoxin to produce hydrogen sulfide. Hydrogen sulfide is also produced through cystathionine beta-synthase and cystathionine gamma-lyase, along with 3-MST, and is known to alleviate a variety of illnesses such as cancer, heart disease, and neurological conditions. The importance of cystathionine beta-synthase and cystathionine gamma-lyase in hydrogen sulfide biogenesis is well-described, but documentation of the 3-MST pathway is limited. This account compiles the current state of knowledge about the role of 3-MST in physiology and pathology. Attempts at targeting the 3-MST pathway for therapeutic benefit are discussed, highlighting the potential of 3-MST as a therapeutic target.

## 1. Introduction

Hydrogen sulfide, a colorless gasotransmitter, is enzymatically regulated by cystathionine β-synthase (CBS), cystathionine γ-lyase (CSE), and 3-mercaptopyruvate sulfurtransferase (3-MST) through direct or indirect use of sulfur-containing amino acids as their substrates [1,2,3], although a small portion of H_2_S is produced via non-enzymatic reactions [4]. The CBS pathway involves the production of H_2_S by catalyzing the condensation of L-cysteine and L-homocysteine. With this pathway, a supplement of L-cystathionine is produced along with H_2_S [5,6]. The primary H_2_S generator in the central nervous system is cytoplasmic CBS [7,8]. The CSE pathway involves the production of H_2_S using L-cysteine as a substrate with a supplement of L-serine [9,10]. CSE is in the cytosol under typical resting conditions and is abundantly expressed in the cardiovascular and respiratory systems [11,12,13]. There are extensive studies documenting the significance of the two enzymes CBS and CSE [6,9,14]; however, the 3-MST pathway is relatively unexplored, and understanding its role in physiology, specifically pathology, could be valuable. This review dives into the role of endogenous 3-MST in pathophysiology.

3-MST is a protein with a rhodanese-like domain, composed of distinct C- and N-domains [15,16]. Rhodanese are enzymes that typically engage in the catalysis of thiosulfate to cyanide by the transfer of a sulfur atom, while 3-MST catalyzes the production of H_2_S with 3-mercaptopyruvate as the donor of sulfur [15,17]. 3-MST is highly expressed in the kidney, liver, brain, testes, and large intestine, and is most prevalent in the endocrine system [18]. Ample amounts of 3-MST mRNA were also detected in the kidney, liver, and heart [19]. Since the mitochondria contain three times as much L-cysteine as the cytoplasm, involvement of 3-MST in H_2_S signaling could be predominant in the mitochondria. The process to form H_2_S from 3-MP by 3-MST involves two different catalytic reactions (Figure 1). More specifically, 3-mercaptopyruvate (3-MP) can be produced either by the cysteine aminotransferase (CAT) pathway or the diamine oxidase (DAO) pathway. CAT catalyzes the transfer of the amino group from L-cysteine to α-ketoglutarate, releasing L-glutamate and yielding 3-MP. DAO, however, transforms D-cysteine. 3-MST catalyzes the release of H_2_S from 3-MP in the presence of thiols such as thioredoxin (Trx) and dihydrolipoic acid (DHLA) [10]. The conformation of 3-MST alters throughout this chemical reaction. 3-MST is a Zn-dependent enzyme that exists in a monomer-dimer equilibrium. It remains in its active form, as a monomer, when it serves as a reactant of the reaction; however, after interaction with 3-MP, Trx, and DHLA, the structure of 3-MST changes into a biological inactive dimer [20,21]. The dimer is created by oxidizing exposed cysteine residues and creating an intersubunit disulfide link. After cysteine is oxidized to low-redox-potential sulfenate, its catalytic activity is diminished. Overall, the cysteine in the catalytic region functions as a redox-sensing molecular switch, maintaining the redox-dependent regulation of 3-MST activity [20,21,22,23]. The expression of 3-MST is modulated by Trx and other thiol small molecules such as 3-MP, L-cysteine, glutathione, and DHLA [10,24]. The modulation of the CAT and DAO pathways also has a vast effect on the generation of H_2_S by 3-MST. The 3-MST pathway is reported to significantly contribute toward H_2_S homeostasis in the brain, kidney, and retinal cells, potentially due to its high expression [25,26,27,28,29]. Mechanistic elucidation has confirmed formation of enzyme-bound hydropersulfide (3-MST-Cys-S-SH) resulting from the catalysis of desulfuration of 3-MP by 3-MST. Transfer of the outer sulfur atom called sulfane sulfur to proteins or small molecule thiol acceptors, resulting in their persulfidation (RS-SH), is deemed responsible for H_2_S signaling mechanisms. A polysulfide species (3-MST-SnSH) is also produced by 3-MST-SSH, and they similarly persulfidate thiols and other proteins [30,31,32,33].

The importance of H_2_S homeostasis in disease has garnered interest in understanding the role of 3-MST, along with CBS and CSE, in cellular oxidative processes as one of the endogenous H_2_S-producing enzymes. In this review, we discuss the significance of 3-MST pathway in the pathophysiology of various diseases and recent advances in the development of therapies targeted toward this pathway. Existing studies on 3-MST primarily focus on its role in cardiovascular function (especially hypertension, angiogenesis, and myocardial injury), cancer (particularly neuroblastoma, colon cancer, and carcinoma), neurology (mainly ischemia and hypoxia), liver function, and prevention of cyanide toxicity; however, some studies briefly mention 3-MST’s role in pathologies such as obesity, kidney injury, adipogenesis, etc. The majority of the studies showed reduced 3-MST levels in diseases such as brain injury and angiogenesis, although a few studies claim that increased expression of 3-MST is the causative factor behind several diseases, including various cancers such as bladder cancer, prostate cancer, etc. A compilation of all pathologies wherein the role of 3-MST has been investigated is included in Table 1, which presents the results of individual experimental studies along with any pathway-specific finding for the specific disease condition. The extent to which 3-MST-mediated release of H_2_S influences disease pathology is still unknown. Hence, focusing on 3-MST will be helpful for further exploration of this pathway in preclinical research by providing the current state of experimental knowledge and conclusive insights regarding the disease-specific contribution of 3-MST in H_2_S homeostasis. 

## 2. Role of 3-MST in Disease

### 2.1. Cancer

The role of 3-MST in various cancers has been extensively studied by researchers across the globe (Figure 2). The studies looked at differential expressions of 3-MST in cancer specimens obtained from in vitro and in vivo experimental conditions. 

An in vitro study conducted in organoids from human colon epithelial cells with driver mutations (NL, A, AT, AKST, AdeCINTS, and AdeCINTSK) showed over-expression of 3-MST [34]. Experiments with AdeCINTS organoids showed that reduction in 3-MST itself is responsible for the increase in the expression of 3-MST. This mechanism was attributed to a self-amplification positive feedback mechanism between production of H_2_S and 3-MST expression. Experiments with HMPSNE, 2-[(4-hydroxy-6-methylpyrimidin-2-yl)sulfanyl]-1-(naphthalen-1-yl)ethan-1-one, and a 3-MST-specific inhibitor demonstrated anti-proliferative effects on different organoids as well as apoptosis of colon cancer cells [34,35]. The findings also suggest that upregulation of 3-MST in cancer cells imparts cytoprotective effects and promotes an aggressive metastatic cancer phenotype [35]. Inhibition of 3-MST exerted a more prominent effect on the regulation of mesenchymal-epithelial transition by stimulation of the antiapoptotic Wnt/β-catenin pathway [34,36]. Higher levels of 3-MST along with H_2_S and other drug-metabolizing enzymes were also found in the chemotherapeutic-agent-resistant HCT116 colon cancer cells [37]. In contrast to these findings, Panza et al. reported lower levels of 3-MST in human urothelial carcinoma cells with very little effect on cell proliferation and apoptosis [38]. Experiments with murine colon carcinoma cell line, CT26, revealed that catalytic activity of 3-MST significantly contributes toward the maintenance of cell migration, proliferation, and regulation of various bioenergetics pathways [39].

Higher levels of 3-MST were found in several other human neoplastic cell lines (neuroblastoma SH-SY5Y, astrocytoma U373, and malignant gliomas U-87 MG, melanoma A375, and WM35) [40,41,42] responsible for sulfane sulfur formation and H_2_S production. A study using antioxidants such as diallyl trisulfide (DATS), an organosulfur compound isolated from garlic, and N-acetyl-L-Cysteine showed increased activity of 3-MST and inhibited cancer cell proliferation in U-87 MG and SH-SY5Y cell lines. As a sulfane sulfur donor, DATS increased the level of sulfane sulfur that gets transferred to the cysteine residue of Bcl-2, an anti-apoptotic protein, resulting in the latter’s inactivation in these cells [40,41]. Expression and activity of 3-MST was also evaluated in other cancer cell lines such as hepatoma cell lines (Hepa1c17, HepG2) [43,44], lung adenocarcinoma lines (A549, H522, H1944) [45,46], renal cell carcinoma lines (RCC4) [47], urothelial cancer cell lines [48,49], and non-small cell lung cancer cell lines (95D) [50]. Higher mitochondrial DNA repair capacity due to H_2_S was observed in cultured lung adenocarcinoma as compared to healthy cells. Inhibiting the activity of H_2_S by HMPSNE hampered the repair capacity of the mtDNA, suppressed mitochondrial bioenergetics, and increased the sensitization of these cells towards chemotherapeutic agents [44]. 

Along with the above discussed in vitro studies, various in vivo studies have substantiated the role of 3-MST in cancer biology. Some of these studies also included the use of inhibitors to depict the specific role of 3-MST in pathways involved in disease progression. Bantzi et al. identified crucial structural alterations in the central core of the inhibitor HMPSNE to enhance the binding affinity to the 3-MST active site (Figure 3) [51]. The lead compounds displayed anti-proliferative effects in MC38 and CT26 colon cancer cell lines and in MC38 tumors developed in immunocompetent mice. The results of this study showed structural motifs that could be used to create 3-MST inhibitors with potent anticancer properties [51]. Additionally, comprehensive analysis of H_2_S-producing enzymes in HCT116 colon cancer cells showed increased levels of 3-MST along with CBS and CSE. Inhibitors of all three H_2_S enzymes and H_2_S donors showed concentration-dependent, biphasic (bell shape) antiproliferative effects on HCT116 colon cancer cells [52]. 

Ex vivo screening of HCC (hepatocellular carcinoma) using iTRAQ-based proteome profiling and miRNA and mRNA profiling identified 3-MST as one of the up-regulated proteins among ten others; 3-MST expression was about three-fold higher in epithelial cell adhesion molecules expressing (EpCAM+) cancer stem cells compared to normal stem cells. Further, transcriptomics analysis confirmed overexpression of 3-MST (two-fold) in EpCAM+ cancer stem cells [53]. In contrast to these findings, Li et al. reported significant suppression of 3-MST mRNA and protein in HCC samples compared to their non-tumor counterparts. As a corollary, 3-MST overexpression mediated inhibition of HCC cell proliferation, enhanced apoptosis, and negatively correlated with the size of tumor xenograft in nude mice [54]. Here, 3-MST-H_2_S mediated down regulation of the cell cycle, and inhibition of the AKT/FOXO3a/Rb signaling pathway was implicated in the suppression of tumor development. 

Single-cell transcriptomic studies carried out in glioblastoma tumors isolated from thirty patients recognized overexpressed 3-MST as a metabolic enzyme playing a crucial role in the motility of glioblastoma cells as well as the development of tumors. Mitochondrial mass load and oxidative stress of these cells were associated with 3-MST mobilization. 3-MST knockdown hampered the glioblastoma cell mobility and integrity, resulting in reduction of tumor burden and a marked improvement in survival of 3-MST knockdown mice. This result was further supported by a study carried out using a pharmacological inhibitor of 3-MST, I3MT3. This effect of 3-MST on GB cells motility was shown to be mediated by the protein persulfidation [55]. 

Increased expression of 3-MST along with CBS and CSE was reported in renal cell carcinomas and was established by a study conducted using 88 human kidney tissue specimens from healthy and cancer patients [56]. Higher expression levels of 3-MST have also been reported in adenoid cystic carcinomas of the oral cavity [57], mouth floor mucoepidermoid carcinoma [58], and oral squamous cell cancer [59]. Contrary to this finding, low expression of 3-MST was found in human melanoma cell lines and tissue samples (from nevi to metastasis) and established CSE’s major role in melanoma [60].

### 2.2. Cardiovascular Disorders

Numerous studies have investigated the role of the 3-MST pathway in various cardiovascular diseases. Limited reports discuss the utilization of in vitro model systems for such purposes. An in vitro study using a human vascular endothelial cell (EC) line showed diminished endothelial cell proliferation and suppressed VEGF-induced EC migration by lentiviral attenuation of 3-MST [61]. Here, experiments with wild-type, shNT, and sh3-MST ECs were used to assess the role of 3-MST in the development of an angiogenic phenotype and observed that the EC tube-like network formation was reduced more in the 3-MST silenced group than either the wild-type or shNT control cells. The study also displayed the role of 3-MST in mitochondrial bioenergetics. Inhibition of 3-MST using I3MT3 lowered mitochondrial ATP generation and EC oxidative phosphorylation. Oxidative stress, however, inhibits 3-MST activity and could have an adverse effect on cellular homeostasis [61]. Oxidative stress in in vitro hyperglycemia or in vivo streptozotocin-induced diabetes inhibited angiogenesis, reduced mitochondrial activity, and delayed wound healing due to diminished 3-MST activity [62]. In contrast, treatment with antioxidants such as DHLA in vitro or DL-α-lipoic acid (LA) in vivo improved endogenous 3-MST function and successfully restored the process of angiogenesis and bioenergetics. The study raises the possibility that a combination of H_2_S donors with 3-MP or lipoic acid could be beneficial in improving angiogenesis and bioenergetics in hyperglycemia [62]. Apart from antioxidants, exercise also induced increased expression of the H_2_S-producing enzymes CSE and 3-MST in aged mice. This reduced the rate of production of oxidative stressors (^•^O_2_ and H_2_O_2_) and increased resistance to the calcium-induced mPTP opening, indicating restoration of mitochondrial function in these mice [63].

Comprehensive in vivo evaluation of the cardiovascular phenotype exhibited by mice lacking 3-MST conducted by Peleli et al. provides direct evidence of the role of 3-MST in cardiovascular physiology [64]. The findings revealed that 3-MST is more abundantly expressed in mice hearts than CSE and CBS. Mice with 3-MST genetic ablation were protected against cardiac ischemia reperfusion injury in young mice. The protective effect of 3-MST deletion was not apparent in aged mice and resulted in hypertension and cardiac hypertrophy due to lower thiosulfate sulfurtransferase expression levels, along with increased cardiac oxidative stress [64]. According to another study, myocardial 3-MST levels were considerably lower in end-stage heart failure patients. Using 3-MST knockout mice, the authors displayed impaired cardiac vascular reactivity and reduced mitochondrial respiration and ATP synthesis in heart failure with a lower ejection fraction. Myocardial metabolomics revealed impaired catabolism of branched-chain amino acids responsible for cardiovascular dysfunction. Deleterious effects of 3-MST deficits in heart failure with lower ejection fractions were ameliorated by pharmacological restoration of myocardial branched-chain amino acid catabolism with 6-dichlorobenzo1 [b]thiophene-2-carboxylic acid (BT2) and delivery of a strong H_2_S donor JK-1 [65]. Additional studies utilizing myocardial injury models in mice confirmed reduced myocardial expression of 3-MST with a concomitant rise in ROS and ER stress proteins (p-PERK, p-eIF2, IRE1, ATF4 and CHOP), which was alleviated by treatment with NaSH [66,67]. 

Further, a study recently revealed that the principal regulator of H_2_S generation and function in the coronary artery is 3-MST rather than CSE. This study was conducted by Kuo et al. and investigated the physiological significance of two H_2_S enzymes, CSE and 3-MST, in coronary vasodilation. While 3-MST was discovered in human coronary artery endothelial cells as well as rat and mouse coronary arteries, CSE was not present in the coronary vasculature. Rat coronary artery homogenates produced H_2_S in vitro in a 3-MST-dependent manner. Additionally, in vivo coronary vasorelaxation responses were also noticed in CSE knockout mice, wild-type mice (WT), and WT animals administered with the CSE inhibitor propargylglycine, demonstrating minimal contribution of CSE-produced H_2_S in coronary vasoregulation [68]. In another study, atrial fibrillation dramatically decreased 3-MST and CSE expression as well as levels of H_2_S, stimulating CBS expression [69].

The importance of the 3-MST-H_2_S pathway in hypertension-induced damage has been studied using rat models. Firstly, H_2_S production in spontaneous hypertensive rats was significantly lower compared to normotensive rats, in agreement with reduced 3-MST cardiac expression. Further, sulfane sulfur levels were significantly lower in the hearts of hypertensive rats than in young and old normotensive rats [70]. Another study showed antioxidant effects of increased 3-MST expression in young hypertensive rats responsible for metabolism of sulfane sulfur and production of H_2_S. The findings indicated that 3-MST, along with rhodanese, contribute significantly to reversing the effects of aging in normotensive rats and young hypertensive rats [71]. In contrast to the above findings, another study depicted the role of the 3-MST/H_2_S pathway in hypertension; in the study, H_2_S levels were found to be higher in the erythrocytes and serum of patients with untreated hypertension. Since CBS and CSE proteins were non-traceable in erythrocytes, it is likely that the 3-MST pathway is primarily responsible for the release of human erythrocyte endogenous H_2_S [72]. 

### 2.3. Neurological Disorders

In addition to CBS and CSE, the importance of 3-MST-derived synthesis of H_2_S in the brain is increasingly being understood. While 3-MST is largely found in neurons, CBS is primarily found in astrocytes. Decreased expression of 3-MST was noted in a dermal fibroblast cellular model of human Down syndrome by Theodora et al. Western blot analysis demonstrated lower expression of 3-MST (but not CSE or CBS) in these cells as compared to controls [73]. In vivo validation of 3-MST’s role in brain function was achieved by genetic ablation of the 3-MST gene in C57BL/6 mice using embryonic stem cells. The prefrontal cortex (PFC) of the 3-MST-KO mice had higher levels of serotonin, which were associated with considerably increased anxiety-like behaviors without any aberrant structural abnormalities in the brain. Antioxidant deficiency, diminished H_2_S signaling cascade, and/or SOx deficiency due to 3-MST deletion were responsible for the aberrant behavior seen in 3-MST-KO mice [20]. Similarly, downregulation of 3-MST along with significant inhibition of endogenous H_2_S generation was apparent in the hippocampus of rats subjected to sleep deprivation-induced cognitive impairment, implying its role in cognitive function [74]. Another study investigated the influence of intragastric daily administration of low-dose melatonin on age-dependent changes in the hippocampus. This multiomics study showed impaired lipid homeostasis responsible for mitochondrial dysfunction and neuronal damage in the aging brain; these symptoms were reversed with melatonin by modulation of key proteins such as 3-MST that are involved in lipid regulation in the hippocampus [75]. 

Another study looked at the contribution of H_2_S toward early blood brain barrier (BBB) disruption and post-ischemic cerebral vasodilation/hyperemia. After inducing transitory focal cerebral ischemia, the use of CSE and CAT/3-MST inhibitors greatly prevented post-ischemic cerebral vasodilation and avoided early BBB disruption, which was evident from the avoidance of fluorescein and Evans blue extravasation [76]. This is contrary to the protective effects exhibited by exogenous H_2_S in this mouse model on late BBB disruption. Another investigation found that ischemic cerebral blood capillaries had much lower amounts of H_2_S and 3-MST than non-ischemic arteries. The resulting impairment in mitochondrial and cerebrovascular endothelium function correlated with blood H_2_S levels and 3-MST expression [77]. Treatment of rats exposed to ischemia/reperfusion injury by betaine prevented brain damage in this animal model [78]. Mechanistic investigation showed enhanced expression of superoxide dismutase 1 (Sod1) and glutathione peroxidase 4 (Gpx4), as well as 3-MST, following pretreatment with betaine. Zhang et al. investigated the role of 3-MST in a traumatic brain injury (TBI) mouse model and discovered that upregulation of 3-MST was associated with neuronal autophagy after TBI insult [79]. The levels of 3-MST peaked following the insult and then steadily decreased until they reached a valley. Importantly, immunohistochemical analysis demonstrated that injury-induced expression of 3-MST partially colabeled with LC3 (autophagy marker), but not with propidium iodide (a cell death marker). These findings revealed that 3-MST, mostly found in live neurons, may be involved in neuronal autophagy and the pathophysiology of the brain following a TBI insult [79]. 

Low expression levels of 3-MST were detected in the peripheral blood mononuclear cells obtained from the patients of multiple sclerosis. The study also noted that there was a significant inverse relationship between the expression of 3-MST and proinflammatory markers, whereas pretreatment with GYY4137 (H_2_S donor) significantly reduced the levels of interferon (IFN)-γ and interleukin (IL)-17 production in the lymph node and spinal cord encephalitogenic T cells [80]. In contrast, the expression of CBS and 3-MST was noticeably greater in cerebrospinal fluid (CSF) samples from patients after subarachnoid hemorrhages (SAH) as well as in rat CSF samples following SAH. Strong associations were found between the rise in IL-6 two days after SAH and the levels of CBS, DAO, and 3-MST; poor outcomes at six months after SAH were linked to high levels of CBS, 3-MST, and DAO in the human CSF samples [81]. 

Our lab explored the role of H_2_S in memory and cognition and we evaluated the involvement of the 3MST/H_2_S pathway in Alzheimer disease (AD) pathology using a symptomatic transgenic AD mouse model (APP/PS1). A significant reduction in 3-MST activity, but not its expression, was observed in the cortex and hippocampal regions of these mice. When administered with a 3-MST substrate analogue, sulfanegen, restoration of brain 3-MST function was confirmed, which resulted in prevention of cognitive impairment and reversal of oxidative and neuroinflammatory consequences of AD pathology. Quantitative neuropathological analyses showed significant disease-modifying effects of the compound on amyloid plaque burden, brain inflammatory markers, and by attenuation of progressive neurodegeneration in these mice, as evident from the restoration of TH+ neurons in the locus coeruleus [82]. In another investigation with APP/PS1 mice, reduced 3-MST protein levels were restored to control levels after NaSH administration. Supplementation with NaSH reduced APP, BACE1, and Aβ_1-42_ protein levels and upregulated nuclear factor erythroid-2-related factor 2 (Nrf2) [83]. These studies suggest supplementation of 3-MST function in the brain as a viable approach for the management of AD and other neurological diseases. 

### 2.4. Cyanide Toxicity

Implications of 3-MST in the detoxification of cyanide and development of effective antidotes has been a topic of military and clinical relevance. Cyanide is one of the fastest-acting hazardous compounds and can cause clinical poisoning in seconds to minutes. Rhodanese (thiosulfate sulfurtransferase, TST) and 3-MST are recognized as the major detoxification pathways for cyanide. Singh et al. studied dose-dependent effects of potassium cyanide (KCN) on cyanide levels and activities of cytochrome c oxidase (CCO), TST, 3-MST, and CSE [84]. At 0.5LD50 dose, increased liver cyanide levels were associated with enhanced liver activities of TST, 3-MST, and CST as well as with inhibition of CCO; however, the elevated renal cyanide corresponded only with increased 3-MST activity in the kidney. Appreciable brain 3-MST activity, but not activity of TST, was also noted in this study [84]. Another study looked at acute effects of sub-lethal doses of cyanide on the activity of 3-MST in the liver, kidney, and several brain areas of mice. In agreement with the previous study, this high dose of cyanide hindered 3-MST’s ability to effectively detoxify cyanide. However, cyanide toxicity could be tolerated due to the abundance of 3-MST in the liver and kidney and the total sulfane sulfur resulting from this pathway [85]. Toxic effects of cyanide are mitigated by 3-MST through transsulfuration, which transforms it into the less toxic thiocyanate (Figure 4). Exclusive localization and activity of TST in the mitochondria and the inability of sulfur donors to effectively permeate mitochondria brings into question TST’s role in cyanide detoxification. 3-MST expression is increased in various tissues upon cyanide exposure and is distributed in both the cytoplasm and the mitochondria, which means that it first detoxifies cyanide in the cytoplasm and the remainder that escapes into mitochondria is neutralized by 3-MST along with TST [19,86]. 

Through the utilization of various mimics of 3-MP, attempts at the modulation of 3-MST activity and the development of cyanide antidotes were made. Compounds such as hypotaurine and methanesulfinic acid, which double 3-MST activity in bovine kidney extract, have been discovered [87], while pyruvate, phenylpyruvate, oxobutyrate, and oxoglutarate have been found to inhibit the enzyme. Another interesting study identified picrylsulphonic acid as an inhibitor of TST, as it enhanced detoxification of cyanide by the 3-MST pathway [88]. This compound could be useful as a chemical tool to investigate the 3-MST-H_2_S pathway in cyanide detoxification. A thiol donor, lipoic acid, was able to reverse toxic effects of cyanide and was shown to enhance 3-MST activity and inhibit peroxidative processes in the kidney [89]. A rational drug design approach was implemented by Patterson et al., resulting in the development of 3-MST-targeted cyanide antidotes [90,91]. The instability of the 3-MST substrate, 3-MP, in blood was addressed in this study by the development of prodrugs and led to identification of a water-soluble antidote, sulfanegen. The efficacy of this compound was evaluated in various models of cyanide toxicity. Brenner et al. examined the efficacy of sulfanegen sodium to reverse cyanide effects in a rabbit model [92]. When given intravenously or intramuscularly, sulfanegen reversed the effects of cyanide exposure on oxyhemoglobin and deoxyhemoglobin substantially and more quickly than in control animals. Additionally, RBC cyanide levels reverted to normal more quickly after receiving sulfanegen sodium. These experiments highlight the utility of the 3-MST pathway for the development of an effective, fast-acting cyanide antidote [92].

### 2.5. Miscellaneous

The role of 3-MST has also been investigated in other disease conditions along with CBS and CSE; thus, direct experimentation specifically targeting 3-MST is limited. The results highlight the importance of 3-MST-mediated H_2_S release in disease progression or pathology, warranting further comprehensive studies.

#### 2.5.1. Obesity/Diabetes

Hu et al. reported that high glucose suppressed the expression level of 3-MST and CSE, hampered the production of H_2_S in 3t3-L1 adipocytes, and promoted activation of ADAM17, an enzyme with pleiotropic effects and contributions in development of T2DM (Type 2 Diabetes Mellitus), angiopathic and ischemic heart disease, chronic kidney disease, etc. The authors concluded that H_2_S-producing enzymes including 3-MST are negatively correlated with the expression of ADAM17. Thus, increasing the H_2_S level and its producing enzymes can be considered a therapeutic intervention for ADAM17-related complications [93]. The effect of 3-MST inhibition by HMPSNE on adipocyte differentiation and lipid accumulation was also investigated, which resulted in their differentiation into mature cells and increased lipid uptake with enhanced inflammation of adipose tissue [94,95]. The possible mechanisms are impaired fatty acid oxidation and oxidative phosphorylation pathways as well as activation of several transcription factors responsible for the cellular differentiation due to inactivation of 3-MST. Thus, targeting 3-MST activity could be considered a promising therapeutic approach to counteract adipose dysfunction-associated conditions. Additional studies evaluating the impact of a high-fat diet (HFD) on 3-MST levels and activity in mice under physiological conditions reported decreased levels of 3-MST in tissues such as skeletal muscle [96], liver [97], and adipose tissue [98,99] and consequently led to disruption of the circadian clock [96], metabolic impairment of the TCA cycle, fatty acid oxidation, oxidative phosphorylation, and lipid accumulation with advancement to obesity [98,99].

#### 2.5.2. Vasculature and Muscles

Given the important vasodilatory effect of H_2_S in numerous vascular beds, studies have been directed at understanding the role of H_2_S-producing enzymes in vascular illnesses. Endothelial H_2_S generation correlated with 3-MST and CSE expression in the cutaneous microvasculature from human participants [100]. Stimulation and inhibition of 3-MST using synthetic compounds altered pulmonary artery vasoconstriction observed during the hypoxic condition, attesting to the role of 3-MST-mediated H_2_S release [101]. Similarly, expression of 3-MST was also detected in human umbilical artery and smooth muscle cells, along with other H_2_S-producing enzymes which enhanced H_2_S-mediated relaxation by activating the potassium channel [102]. In another study, suppression of DAO/3-MST and CAT/3-MST pathways with decreased production of H_2_S were observed and related to erectile dysfunction in diabetic rats [103]. 3-MST and other enzymes mediated release of H_2_S was found to induce muscle relaxation of bladder trips of humans and Sprague-Dawley rats [104], porcine bronchioles from peripheral airway [105], and mice corpus cavernosum [106]. According to these studies, endogenously produced H_2_S by 3-MST and other enzymes exerts its vasorelaxation effect by activating K + ATP channel present on muscle cell membrane, which is further confirmed by using its inhibitor glibenclamide [103,106]. 

#### 2.5.3. Osteoarthritis

Expression of 3-MST was also determined in cartilage of patients with osteoarthritis (OA), 3-MST knockout, as well as wild-type mice with meniscectomized knees and sham-operated mice [107]. Lower levels of 3-MST due to oxidative stress were observed in the cartilage of OA patients and the meniscectomized knees of wild-type mice. Additionally, calcification analysis of 3-MST knockout mice with meniscectomy displayed higher amounts of calcium-containing crystals deposited on joints and cartilage with greater severity of OA as compared to wild-type mice. 3-MST imparts its protective effect by producing H_2_S and decreasing the mineralization and interleukin-6 release in chondrocytes, which are important in OA progression [107]. Higher expression of 3-MST along with other H_2_S-producing enzymes was observed in tissues isolated from a rabbit model of mandibular distraction osteogenesis (DO: a technique with clinical implications in bone reconstruction, maxillofacial deformity correction, limb lengthening etc.) [108]. Progressive down-regulation of H_2_S biosynthetic enzymes was noted during a consolidated duration. Results from this study indicated that the H_2_S signaling pathway is involved in DO by inducing angiogenesis and osteogenesis processes.

#### 2.5.4. Gastrointestinal System

In an acute pancreatitis animal model, decreased activity of 3-MST resulted in increased levels of cysteine, as 3-MST catalyzes the degradation of cysteine [109]. Nagahara et al. also depicted 3-MST as a redox condition dependent, pleiotropic, and housekeeping enzyme [109,110]. 3-MST-mediated synthesis of H_2_S also succeeded in providing protection to gastric mucosa of rats from damage caused by aspirin [111], alendronate [112], and mucosa immobilized with cold water [113]. The role of 3-MST in human ulcerative colitis, Crohn’s disease, and animal models was evaluated. Very low expression of 3-MST was observed in colonic samples with a marked increase in the expression of inflammatory cytokines and apoptotic factors along with ROS markers [114]. 3-MST exerts its protective effect by reducing AKT phosphorylation and increasing AKT expression, which regulates the apoptosis process and protects intestinal epithelial cells from inflammatory mediators. 

Decreased levels and activity of H_2_S and 3-MST were reported in tissues samples from mice with acute kidney injury. Treatment with NaSH successfully reversed the process and protected the kidney by inhibiting oxidative stress and inflammatory response via TLR4/NLRP3 signaling pathways. Thus, in endogenous H_2_S produced by 3-MST, CSE and CBS are key components and are involved in maintaining kidney homeostasis during pathological conditions [115].

#### 2.5.5. Liver Injury

Studies understanding the role of 3-MST in liver diseases are limited. A study explored the contribution of the 3-MST-H_2_S pathway in non-alcoholic fatty liver disease (NAFLD) pathogenesis and in regulation of free fatty acid levels (FFA). Hepatocytes exposed to FFAs in vitro and mice with NAFLD administered with a high fat diet (HFD) showed increased 3-MST expression, which was in part dependent on NF-KB/p65 [116]. Contrarily, another study with healthy mice fed with HFD showed decreased levels of liver 3-MST [97]. Partial 3-MST knockdown by short hairpin RNA or heterozygous 3-MST gene deletion dramatically reduced hepatic steatosis. Co-immunoprecipitation experiments showed that 3-MST directly interacted with and adversely regulated liver CSE. Inhibiting CSE/H_2_S and consequent elevation of the sterol regulatory element-binding protein 1c pathway, C-Jun N-terminal kinase activation, and hepatic oxidative stress were the main mechanisms by which 3-MST induced steatosis [116]. Chronic exercise training significantly increased the levels of H_2_S in the blood and the livers of the mice that were given the HFD by upregulating mRNA expression of CBS, CSE, and 3-MST [117]. Targeting 3-MST may be a potential strategy for treating liver ailments, which indicates significant relevance for 3-MST in pathologies associated to the liver.

#### 2.5.6. Microbes

The role of 3-MST is also evaluated in different microbes. In a study determining how bacteria thrive under oxidative stress conditions, 3-MST was recognized as one of the proteins regulated by MsrR, a multiple stress resistance regulator working as a thiol-based redox sensor during oxidative stress conditions. This was further confirmed by using the 3-MST mutant strain *Corynebacterium glutamicum*, which showed diminished growth under different stress conditions [118]. A similar study demonstrated that H_2_S and nitric oxide work together to support growth in different bacterial species such as *Bacillus anthracis*, *Pseudomonas aeruginosa*, *Staphylococcus aureus*, and *Escherichia coli* whereas inactivation of 3-MST decreases the production of H_2_S, making these pathogens highly sensitive toward antibiotic treatment [119]. In another case, 3-MST-mediated H_2_S production was recognized for improving the lifespan of *Caenorhabditis elegans* [1]. In some in vitro studies, 3-MST is also recognized as a potential selenium transporter which carries selenium in the form of selenodiglutathione in metabolic pathways, where selenophosphate synthesis is required to produce Secys-tRNA, a precursor of selenocysteine present in selenoenzymes in bacteria and mammals [120]. Contrarily, supplementation of selenium deficient diets with chicken resulted in abnormal increases in the levels of H_2_S released by 3-MST, CSE, and CBE, inducing liver injury [121,122]. These studies highlight the importance of achieving proper balance between selenium supply and the H_2_S-producing role of 3-MST for desired pharmacological effect. 

## 3. Conclusions and Future Directions

Despite the fact that many in vivo and in vitro studies have been conducted, understanding of how 3-MST-mediated release of H_2_S affects different pathological conditions remains limited. Some studies suggest increasing 3-MST expression to slow the progression of the disease, while others suggest blocking the 3-MST pathway as a treatment. There is thus a need for systematic probing of the intricate biochemical mechanisms highlighting the specific role of the 3-MST/H_2_S pathway in disease pathology. Finally, creating H_2_S donors or compounds that improve 3-MST expression and function, or specifically inhibiting 3-MST as a therapeutic intervention, could be beneficial. Novel combinatorial approaches such as the combination of H_2_S donors or antioxidants with 3-MP or its substitutes could be explored to effectively utilize this pathway for therapeutic benefit. There have been great efforts to utilize triple gene therapy consisting of CBS, CSE, and 3-MST in mice with hyperhomocystemia [123]. Increasing protein expression of these three H_2_S biosynthetic enzymes proved beneficial in catalyzing the conversion of homocysteine to H_2_S and reducing its level in cells and tissues. This therapy also succeeded in providing protection to the mitochondria of mouse aortic endothelial cells by attenuating the homocysteine-induced expression of LC3I/II, a mitophagy marker [123]. Along with gene and protein therapy approaches, rational drug design methods for the development of 3-MST-specific substrates or inhibitors need further experimentation for therapeutic applications. Chemical inhibitors of the 3-MST enzyme, in addition to those described in Section 2.1 obtained via rational inhibitor design, were also discovered through a high throughput screening approach [124]. The identified lead compounds offered selectivity for 3-MST over other H_2_S-producing enzymes and could be valuable tools to study the physiological role of 3-MST. The study suggested the significance of sulfane sulfur produced by 3-MST in the physiological role of this enzymatic pathway that was previously thought to be regulated by H_2_S. This review could serve as a basis for further consideration of 3-MST as a potential pharmacological target.

**Table 1 antioxidants-12-00603-t001:** Studies related to 3-MST mediated H_2_S release in disease pathology.

	Model Used	Observation/Findings
Cancer		
**Colon cancer**	Mutant organoids of human intestinal epithelium [34], HCT116, HT29, and Lovo cells [36,52,125] 5-Flourouracil resistant HCT116 [37], CT26 murine carcinoma cells [39] Colon cancer tissues patients [52]	Upregulation of 3-MST/H_2_S pathway at early stage of cancer development [125]. H_2_S produced by 3-MST and other H_2_S-producing enzymes activated the CyR61 promoter by sulfhydration of Sp1 (CyR61 activator) [35]. Increased level of 3-MST also displayed cancer cell protective effect 1 [37,39,52]. Increased 3-MST expression contributed to generation of H_2_S in 5-FU-resistant cells [37] and regulated the CT26 cell migration, proliferation, and bioenergetics in CT26 cells [39]. Endogenous H_2_S produced by 3-MST promoted the epithelial-to-mesenchymal transition (EMT) by enhancing the ATP citrate lyse (ACLY) expression involved in Wnt-β-catenin pathways [36].
**Hepatocellular carcinoma (HCC)**	Liver tissues of patients with HCC [53], HCC cell lines, LM3 xenografts mice model [116]	Increased level of 3-MST in epithelial cell adhesion molecule containing cancerous stem cells isolated from HCC patients [53]. Low tumor growth rate was observed in HCC xenograft mice overexpressing 3-MST [116]. 3-MST overexpression greatly reduced cell proliferation and growth by triggering G1-phase cell cycle arrest and controlling the AKT/FOXO3a/Rb pathway in HCC cells [116].
**Renal cancer**	Tissue samples from human patients [56], T24 and UMUC3 urothelial cancer cell lines [38]	Differential expression of 3-MST and other H_2_S-producing enzymes observed irrespective of renal cancer metastasis, size, grade, and recurrence [56]. Low level of 3-MST was observed in urothelial cancerous cell line as compared to normal cells [38].
**Glioblastoma (GB)**	GB patients derived cells (PDC), mice xenograft model of small hairpin RNA-induced knockdown 3-MST PDC and shControl PDS [55]	3-MST knockdown in PDC using small hairpin RNA impaired cell’s motility, shape, and invasion ability, resulting in less tumor burden and higher survival observed in mice xenografted with 3-MST deleted PDC compared to shControl PDS [55]. 3-MST-mediated protein persulfidation required to protect the cells from hyperoxidation [55].
**Human neoplastic cancer cells**	Astrocytoma U373 cells, neuroblastoma SHSY5Y cells, melanoma, melanoma WM35 cells, A375 from solid metastatic cancer [40,42,60]	High 3-MST expression was observed in all cell lines. 3-MST also showed better activity than CSE in these cell lines, thus it was also considered as a major protein involved in sulfane sulfur production. H_2_S produced from CSE induced proapoptotic effects on human melanoma by reducing the activation of ERK/pERK and Apk/pApk pathways and by inhibiting NF-kB mediated anti-apoptotic genes [60].
**Lung adeno-carcinoma**	Human lung adenocarcinoma tissues, mice xenograft model of human lung cancer developed using A549 cells [46]	Combinatorial therapy of H_2_S-producing enzyme inhibitors and chemotherapeutic agents had a greater beneficial effect in lung cancer. Inhibition of 3-MST along with other H_2_S producing enzymes impaired mitochondrial bioenergetics and decreased mitochondrial DNA repair capacity [46].
**Oral cancer**	Tumor biopsies from ACC patients [57], biopsies from Mucoepidermoid Carcinoma (MEC) patients [58], biopsies from Oral Squamous cell carcinoma (OSCC) patients [59]	Levels of 3-MST and other H_2_S producing enzymes were increased more in human ACC [57], MEC [58], and OSCC [59] than adjacent benign tissues.
**Cardiovascular Disorders**		
**Angiogenesis**	bEnd3 cells and male Sprague-Dawley rats [62]	Increased H_2_S production in bEnd3 cells and plasma H_2_S levels in rats reduced 3-MST activity [62]. Hyperglycemia impaired 3-MP/3-MST/H_2_S pathway and mitochondrial function; proangiogenic effect of 3-MP in vitro was associated with the activation of Akt and Protein Kinase G (PKG) [62].
**Cardiac injury**	Male Sprague-Dawley rats [126]	Lower 3-MST levels significantly increased NADPH Oxidase 4 (NOX4) and p67 protein expressions in cardiac injury [126]. H_2_S had cardio-protective effects via decreasing NADPH oxidase and ROS production.
**Heart failure**	Myocardial samples from patients and 3-MST knockout and wild-type mice subjected to acute heart failure [65], SD rats induced with Angiotensin-II and Left atrial appendage (LAA) tissue collected from rheumatic heart disease (RHD) patients [69]	Reduced 3-MST levels observed in failing patients [65]; reduced 3-MST and H_2_S levels in RHD patients [69]. Induction of heart failure in 3-MST KO mice led to poor exercise performance due to increased branched-chain amino acid accumulation in the myocardium, which was linked to decreased mitochondrial respiration, ATP synthesis, and exacerbated cardiac and vascular dysfunction [65]. Atrial Fibrillation reduced 3-MST expression and H_2_S level, increased ERS and atrial fibrosis, and promote left atrial dysfunction in SD rats [69].
**Hypertension**	Male Wistar-Kyoto rats [70], blood samples collected from hypertensive patients and normotensive patients [72]	Lower expression and reduced 3-MST activity was observed in old hypertensive rats compared to young hypertensive [70]; erythrocyte and serum H_2_S levels were higher [72]; CBS and CSE levels were not detected in erythrocytes. Thus, 3-MST/H_2_S pathway is activated in hypertensive patients [72]
**Myocardial infarction**	Sepsis model was induced in Sprague-Dawley rats by cecal ligation and puncture (CLP) [66]; primary cultures of neonatal cardiomyocytes and adult male C57BL/6 mice [67]	Levels of 3-MST were reduced in the sepsis model [66]. Reduced plasma H_2_S levels corresponded with increased expression of endoplasmic reticulum stress marker proteins, including p-PERK, p-eIF2, IRE1α, ATF4, and CHOP [66]. Myocardial infarction surgery decreased 3-MST levels [67].
**Neurodegenerative Diseases**		
**Acute stroke**	Permanent occlusion of the left middle cerebral artery was induced in the SD rats [127]	Downregulation of 3-MST in both cortex and striatum [127].
**Alzheimer’s disease**	Male APPswe/PS1dE9 AD mice and matched wild-type WT (C57B6) mice [83], SHSY-5Y cells, and APP/PS1 mice [82]	Reduced 3-MST expression [83]; reduced 3-MST activity and 3-MP levels [82] in APP/PS1 mice brain. Increased APP, BACE-1, and Aβ42 levels were observed in APP/PS1 mice compared to WT mice; however, NaSH treatment led to activation of Nrf2/ARE pathway [83]. Attenuation of neuroinflammation (TNFα, IL-6), elevated Aβ42 levels and oxidative stress in APP/PS1 mice by 3-MP prodrug, sulfanegen with restoration of cognitive impairment [82].
**Anxiety-like behaviors**	3-MST KO mice using C57BL/6 embryonic stem cells [20]	MST KO mice showed increased anxiety-like behavior and increased 5-hydroxyindoleacetic acid (5-HIAA) and 5-hydroxytryptamine (5-HT) levels [20].
**Down’s syndrome**	Human dermal fibroblasts [73]	Increased 3-MST levels were found in human Down syndrome fibroblasts compared to controls [73]. Pharmacological suppression of 3-MST activity increased cell proliferation and mitochondrial electron transport and oxidative phosphorylation [73].
**Ischemia/reperfusion injury**	PC-12 cells and male SD rats [78]	Betaine attenuates oxidative stress damage in vitro and I/R induced brain damage [78]. Increased inflammatory markers (IL-1β, IL-6 and TNFα), glutathione peroxidase 4 (Gpx4), superoxide dismutase 1 (Sod1), and 3-MST expression levels after I/R injury were reversed by betaine treatment [78].
**Multiple sclerosis (MS)**	C57BL/6 mice femurs were used for the isolation of bone marrow cells and peripheral blood mononuclear cells (PBMC) obtained from MS patients [80]	Lower 3-MST function in PBMC from MS patients [80]. The expression of 3-MST and pro-inflammatory markers showed a significant inverse correlation [80].
**Hypoxia/oxygen-glucose deprivation**	Primary brain vascular endothelial cells and SD rats [77]	OGD/R induced reduction in H_2_S and 3-MST levels in both ECs and mitochondria also enhanced oxidative stress. Cellular oxidative stress; reduction in mitochondrial potential and ATP levels/ATP synthase activity in hypoxia, which were ameliorated by 3-MP by inhibition of RhoA/ROCK pathway [77].
**Schizophrenia**	B6 (C57BL6/NCrj) and C3H (C3H/HeNCrj) mice were used [128]	Proteomic analysis of brain in these strains showed elevated levels of 3-MST, H_2_S polysulfide-producing enzyme, and greater sulfide deposition in C3H than B6 mice [128]. 3-MST-Tg mice showed reduced ATP levels and decreased ATP-to-ADP ratio, and deficits in cytochrome c oxidase activity, compared to the non-Tg animals [128].
**Sleep deprivation**	Adult male Wistar rats treated with 72 h sleep deprivation (SD) [74]	3-MST levels in the hippocampus of SD-treated rats were reduced [74], triggering increase in autophagosomes, beclin-1 and LC3 II/LC3 I, and down-regulation of p62 [74].
**Subarachnoid hemorrhage**	Human CSF samples and SD rats [81]	Increased 3-MST levels in human CSF samples after SAH [81], displaying correlations between increase in 3-MST and IL-6 [81].
**Traumatic brain injury**	Adult male CD1 mice subjected to TBI [79]	Time-dependent increase in the levels of 3-MST reached peak after first day of injury and reached valley on the third day [79]. Upregulation of 3-MST in the brain cortex was associated with the neuronal autophagic protective effect after TBI, as 3-MST-expressing neurons partially displayed LC3 positive [79].
**Cyanide Toxicity**		
**Cyanide toxicity**	Human blood samples, HEK and A549 [129], Balb/co mice [85], pathogen-free white rabbits [92]	Three different polymorphisms with rare Tyr85 mutation were observed in human blood sample [129]. Individuals with non-sense mutation Tyr85 of 3-MST were more prone to develop cyanide-induced neurotoxicity [129]. 3-MST played a major role in cyanide detoxification in liver and kidney [85], and cyanide levels in erythrocytes, deoxyhemoglobin and oxyhemoglobin [92]. L-cysteine in the presence of 3-MST produced sulfane sulfur that was transferred by rhodanese to detoxify CN-forming SCN [85].

## Figures and Tables

**Figure 1 antioxidants-12-00603-f001:**
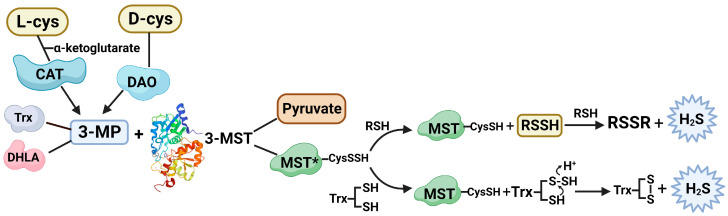
Physiological reaction of 3-mercaptopyruvate sulfurtransferase (3-MST) to produce H_2_S from substrate 3-mercaptopyruvate (3-MP) in the presence of endogenous cofactors thioredoxin (Trx) and dihydrolipoic acid (DHLA). 3-MP is generated from L-cysteine and D-cysteine by cysteine aminotransferase (CAT) and diamine oxidase (DAO), respectively. Reaction of 3-MP with 3-MST causes persulfidation of cysteine sulfhydryl, which sequentially reacts with thiol-containing substrates depicted as R-SH, resulting in the release of H2S. Similarly, reaction of reduced thioredoxin with persulfidated 3-MST yields oxidized Trx with release of H_2_S. Created with Biorender.com (accessed on 18 January 2023).

**Figure 2 antioxidants-12-00603-f002:**
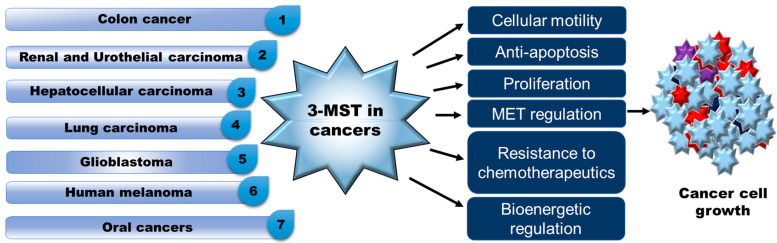
Role of 3-MST in cancer. 3-MST: 3-Mercaptopyruvate, MET: Mesenchymal epithelial transition. 3-MST directly or indirectly plays its critical role in various cancer conditions by regulating pathways involved in the progression of cancers. Pleotropic role 3-MST is observed in different mechanisms regulating cell proliferation and cell death in different cancers.

**Figure 3 antioxidants-12-00603-f003:**
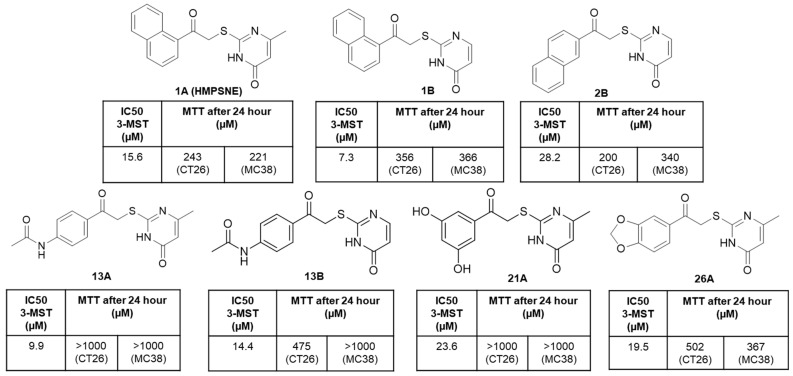
Recently developed 3-MST specific inhibitors and characterization for the anti-cancer activity. Compounds were subjected to IC50 determination in enzymatic assay with recombinant 3-MST (r3-MST) and in cell culture cytotoxicity assays using mouse CT26 and MC38 colon cancer cell lines. Compound 1B exhibited higher rate of inhibition against the enzyme in this series of compounds. Created using data from Bantzi et al. [50].

**Figure 4 antioxidants-12-00603-f004:**
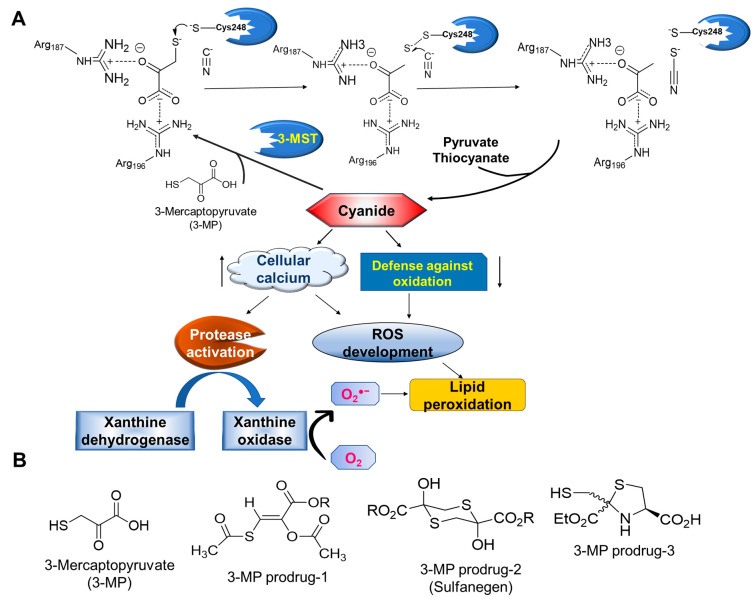
Graphical representation of cyanide-induced toxicity and mechanistic depiction of 3-MST-mediated detoxification of cyanide. (**A**) Oxidative stress resulting from cyanide toxicity is manifested by inhibition of antioxidant enzymes and increased intracellular calcium levels leading to superoxide formation. Detoxification catalyzed by 3-MST involves transfer of sulfane sulfur resulting in nontoxic thiocyanate. (**B**) Chemical structures of 3-MST-targeted prodrugs synthesized as potential cyanide antidotes.

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
