# Peer review of "Role of 3-Mercaptopyruvate Sulfurtransferase (3-MST) in Physiology and Disease"

_antioxidants, 2023, doi:10.3390/antiox12030603_

Round 1

Reviewer 1 Report

This review is an impressive compilation of the literature on 3-Mercaptopyruvate Sulfurtransferase (3-MST). The emphasis lies on the association of 3-MST with various diseases. Therefore, this review is structured according to diseases and the major findings of the studies related to them are summarized. This leads to a very condensed information that may overwhelm the general reader.

Other recent reviews on this topic do not exist. This review is up to date and cites predominantly publications published during the last decade.

Reviewer 2 Report

It is an interesting review article on the production of hydrogen sulfide by 3-MST and its physiology with relevant diseases.

1. Many review articles are found in the references. Original articles are better to be cited.

2. Line 397. Nagahar is to be revised to Nagahara.

Reviewer 3 Report

The manuscript is very interesting and well presented.

Some figures are not very clear.

Also in introduction the innovation should be clearly stated.

Reviewer 4 Report

The review of Rao et al., describes in detail an important role of 3-MST in normal physiology and different diseases in humans. This is a very due paper because practically all similar reviews concentrate on the role of Cbs and Cse H2S-producing genes under normal conditions and in various pathologies. The present review represents a comprehensive analysis of various data describing the molecular aspects of 3-MST enzyme synthesis and localization and its role in cancer, heart diseases, various neurological conditions, cyanide toxicity etc. Importantly, the authors of this review are well-known specialists in the field. Thus, they studied the involvement of 3MST/H2S pathway in AD using a mouse model.  

Minor Remarks and suggestions:

1. The authors briefly described the protective role of H2S-MST in different microbes but did not even mention a very important role of H2S being a universal defense against antibiotics in bacteria, that contain different genes providing H2S production (MST or Cbs, Cse). E.g. Shatalin et al., 2011, 2021.

2. The introduction reads, "The kidneys, liver, and heart contain sufficient amounts of 3-MST mRNA, while other regions, such as the brain, contain less."-Line 40.

Mst is now shown to be highly expressed in the brain. (Tomita M, Nagahara N, Expression of 3-Mercaptopyruvate Sulfurtransferase in the Mouse. Molecules. 2016). In lines 59-60, the authors themselves confirm this fact.

3. There are misprints: Mingyang et al. investigated the role of 3-MST -260 line really Zhang, M.;

4. Line 250- The use of CSE and CAT/3-MST inhibitors greatly prevented post-ischemic cerebral vasodilation and avoided early BBB disruption as evident from avoidance of fluorescein and evans blue extravasation after transitory focal cerebral ischemia- not clear

5. No transcription of abbreviations lines 250-BBB; 274 –CSF, 346-T2DM

Reviewer 5 Report

This is a review article by experienced group that address the relevance of 3MST to neurodegeneration and cognitive impairment observed in AD (81). In this review article, the authors provide the role of 3-MST in diseases in cancer, cardiovascular disorders, and cyanide toxicity, besides neurological disorders. The contents also include the role of 3-MST in miscellaneous disorders such as obesity/diabetes, vascular/gastrointestinal illnesses, osteoarthritis, and liver Injury.

This review article provides important information and suggestions in that the specific role of the 3-MST/H2S pathway need to be examined according to the disease pathology. The review has elements that are organized in a meaningful manner to inform the reader of roles of 3-MST/H2S pathway in various tissues and organs other than the brain. As the authors state, the review could serve as a basis for further consideration of 3-MST as a potential therapeutic target.

I have only a concern to address as below.

The authors would better off mentioning the current status of selective 3MST inhibitor development (Discovery and Mechanistic Characterization of Selective Inhibitors of H2S-producing Enzyme: 3-Mercaptopyruvate Sulfurtransferase (3MST) Targeting Active-site Cysteine Persulfide. Sci Rep. 2017 Jan 12;7:40227. doi: 10.1038/srep40227.).
